# Influence of the Ethanol Content of Adduct on the Comonomer Incorporation of Related Ziegler–Natta Catalysts in Propylene (Co)polymerizations

**DOI:** 10.3390/polym15234476

**Published:** 2023-11-21

**Authors:** Mohammadreza Mehdizadeh, Fereshteh Karkhaneh, Mehdi Nekoomanesh, Samahe Sadjadi, Mehrsa Emami, HamidReza Teimoury, Mehrdad Salimi, Miquel Solà, Albert Poater, Naeimeh Bahri-Laleh, Sergio Posada-Pérez

**Affiliations:** 1Iran Polymer and Petrochemical Institute (IPPI), Tehran 14965/115, Iran; m.mehdizadeh@ippi.ac.ir (M.M.); m.nekoomanesh@ippi.ac.ir (M.N.); s.sadjadi@ippi.ac.ir (S.S.); m.emami@ippi.ac.ir (M.E.); 2Research & Development Center, Kermanshah Polymer Company, Kermanshah 14965/115, Iran; teimouryhamid@gmail.com (H.T.); mehrdadslm1987@gmail.com (M.S.); 3Institut de Química Computacional i Catàlisi and Departament de Química, Universitat de Girona, c/ Maria Aurèlia Capmany 69, 17003 Girona, Spain; miquel.sola@udg.edu

**Keywords:** polypropylene, Ziegler–Natta, adduct, ethanol, comonomer, structure properties relationship

## Abstract

The aim of this work is to investigate the influence of the ethanol content of adducts on the catalytic behavior of related Ziegler–Natta (ZN) catalysts in propylene homo- and copolymerizations (with 1-hexene comonomer) in terms of activity, isotacticity, H_2_ response, and comonomer incorporation. For this purpose, three MgCl_2_.nEtOH adducts with n values of 0.7, 1.2, and 2.8 were synthesized and used in the synthesis of related ZN catalysts. The catalysts were thoroughly characterized using XRD, BET, SEM, EDX, N_2_ adsorption–desorption, and DFT techniques. Additionally, the microstructure of the synthesized (co)polymers was distinguished via DSC, SSA, and TREF techniques. Their activity was found to enhance with the adduct’s ethanol content in both homo- and copolymerization experiments, and the increase was more pronounced in homopolymerization reactions in the absence of H_2_. Furthermore, the catalyst with the highest ethanol content provided a copolymer with a lower isotacticity index, a shorter meso sequence length, and a more uniform distribution of comonomer within the chains. These results were attributed to the higher total surface area and Ti content of the corresponding catalyst, as well as its lower average pore diameter, a larger proportion of large pores compared to the other two catalysts, and its spherical open bud morphology. It affirms the importance of catalyst/support ethanol-content control during the preparation process. Then, molecular simulation was employed to shed light on the iso-specificity of the polypropylene produced via synthesized catalysts.

## 1. Introduction

Currently, polyolefins represent a notable part of the demand for thermoplastics, and the commercial turnover of polyolefin plastics is USD 300 billion per year, with a yearly production of more than 180 million tons. Among the most common polyolefins are polyethylene and polypropylene, with a worldwide annual production volume of 31.2 and 22.7% of the global polymer market, respectively [1].

Due to the industrial application of these thermoplastics, they have always been of interest to both academics and manufacturing companies. Consequently, numerous studies have already been carried out to expedite fundamental reforms on the determining parameters of polyolefins’ microstructures, such as tacticity, molar mass and its distribution, and comonomer incorporation [2,3]. These studies have resulted in the development of various types of modified polyolefin catalysts with the aim of producing advanced polyolefins [4,5]. 

In spite of technological advances in the release of modern coordination catalysts, Ziegler–Natta (ZN)-based formulations still constitute the highest share of the polyolefin catalyst market [1]. Indeed, among the proposed catalyst systems, MgCl_2_/TiCl_4_ is the most advanced type of catalyst in industrial applications for the production of high-density polyethylene (HDPE) [6] and isotactic polypropylene (iPP), with a number of applications [7,8,9]. However, since anhydrous MgCl_2_ has a low surface area and high crystallinity, it is not a suitable carrier for the manufacture of ZN catalysts. Therefore, α-MgCl_2_, as the most abundant crystalline form, is reacted with a Lewis base (mainly ethanol) to improve its characteristics [10]. Therefore, the principal method for preparing spherical ZN catalysts is to use an adduct product with the formula MgCl_2_.nEtOH, where n can be varied from 1 to 6 [11]. By removing the excess alcohol from spherical magnesium chloride, superior porosity can be developed in the adduct texture, enhancing its surface area as a result [12]. Specifically, the effect of the addition of diisobutyl phthalate (DIBP) on the structure of magnesium chloride in the MgCl_2_.EtOH form was assessed, and then the removal of EtOH content was also considered.

Besides surface area, the crystalline structure of neat MgCl_2_ is amended by its modification with ethanol. In this respect, it was well-accepted that the population of the (003) lateral cut, known as the inactive MgCl_2_ surface for TiCl_4_ impregnation, is diminished during adduct and catalyst synthesis [13]. As a consequence, the portion of recognizable surfaces in the MgCl_2_ crystalline structure varies in favor of the fixation of TiCl_4_. 

In previous studies, the influence of the alcohol content on the adduct’s microstructure has been investigated, and divergent results have been reported so far [14]. According to the published literature, with increasing alcohol content, the catalytic activity [14] and tacticity characteristics of the iPPs increase to some extent [15,16]. Despite several works on the aforementioned subject, reports on the comonomer incorporation of ZN catalysts due to the variation of the alcohol content in propylene polymerizations are rare [17].

It is agreed that propylene/ethylene is one of the most used combinations in the polyolefins field, but for industry, the combination with 1-hexene is especially remarkable. Essentially, the alkyl chain of the long α-olefin comonomer improves the final polymer’s mechanical properties. On the other hand, in ZN-assisted olefin polymerizations, the incorporation of α-olefin is a major challenge [18], which suppresses the performance of these catalysts. Improving the related catalyst’s features to successfully incorporate long α-olefins has always been a focus for researchers. In this regard, we have centered our work on the ethanol content of the adduct to determine its effect on the incorporation of 1-hexene as a long alkyl side chain-containing comonomer.

In this study, the effect of the alcohol content (n = 0.7, 1.2, and 2.8) of the MgCl_2_.nEtOH adducts on the related ZN catalyst performance is investigated in relation to copolymerizations [19,20], and, in particular, propylene/1-hexene copolymerizations [18,21]. For this purpose, their activity, comonomer susceptibility toward 1-hexene comonomer in the presence of hydrogen, and their effect on the final structure of the produced polymer are assessed in depth.

## 2. Experimental Materials

Three types of magnesium chloride/ethanol adducts with different percentages of alcohol were prepared from Lorestan Petrochemical Co, Lorestan, Iran, and stored in a dry and moisture-free environment. TiCl_4_ and triethylaluminum (TEAL) were obtained from Sigma Aldrich, Steinheim am Albuch, Germany. DIBP and dimethoxymethylcyclohexylsilane (C-Donor) were used as internal and external electron donors, respectively, and were provided by Merck Co, Darmstadt, Germany. Nitrogen and hydrogen 99.99% were obtained from Roham Gas Co, Tehran, Iran. Propylene gas (polymerization grade) was supplied from Shazand Petrochemical Co, Tehran, Iran. Toluene, heptane, and hexane solvents were also purchased from Sigma Aldrich, Germany, and they were refluxed in the presence of calcium hydrate and Na wire consecutively for 48 h and then placed in a container containing activated molecular sieves and sodium wire.

## 3. Catalyst Synthesis Procedure

Three MgCl_2_.nEtOH adducts with different n of 0.7, 1.2, and 2.8 (Add-A, Add-B, and Add-C, respectively) were employed to prepare the corresponding ZN catalysts according to the procedure as already reported [14]. In a typical procedure, 2.0 g of adduct was placed in a 500 mL glass balloon containing 50 mL of dry toluene under a nitrogen atmosphere at −10 °C. In this study, 50 mL of TiCl_4_ was added gradually for 0.5 h to the suspension. The temperature was raised to 60 °C with a heating rate of 2 °C/min. DIBP (internal donor, 0.14 mL, DIBP/Mg = 0.12 mol/mol) was added to the balloon, then the temperature was increased to 100 °C with the heating rate of 4 °C/min, and it was kept in this condition for 2 h. The supernatant was removed through the use of the dipping technique, and then 50 mL of TiCl_4_ was added together with 30 mL of toluene; the temperature was held at 120 °C for 2 h. The supernatant was removed by dipping the solution over the reactor. The obtained catalyst was washed with dry hexane 3 times at 70 °C, which was subsequently dried and stored under a nitrogen atmosphere.

## 4. Propylene Polymerization

Propylene polymerizations were conducted according to the procedure already reported in our recent work [18]. In the polymerizations, Al/Ti = 180 mol/mol, prepolymerization time = 10 min, prepolymerization pressure = 2 bar, prepolymerization temperature = 35 °C, TEAL/C-Donor = 20 mol/mol, polymerization temperature = 70 °C, polymerization pressure = 6 bar, 5 ml of 1-hexene, polymerization time = 50 min [22].

## 5. Computational Details

Gaussian 16 code [23] was selected to carry out the molecular density functional simulations with the BP86 functional of Becke and Perdew [24,25]. Dispersion corrections were added by means of the method formulated by Grimme (GD3 keyword in Gaussian) [26]. The triple-ζ basis set with the polarization of Ahlrichs (def2-TZVP keyword in Gaussian) [27,28] was selected to describe the electronic configuration of all the systems computed. The frequency calculations were carried out to characterize the minima structures (no imaginary frequencies) and the transition states (one negative frequency that corresponds to the C-C bond formation). 

## 6. Results and Discussion

To investigate the role of the ethanol content of the primary adducts on the performance of the corresponding ZN catalyst in propylene (co)polymerizations [29], three catalysts were synthesized starting from different adduct compositions but similar particle sizes. It is worth mentioning that catalyst particle size originates from adducts particle size, which itself is correlated to the solid content during melt quenching, the stirred speed, and emulsifier content [30]. 

The structure of the catalysts and adducts, including their atomic and crystal structure, configuration, morphology, and particle size characteristics, were thoroughly characterized using different techniques. Then, their performance in propylene polymerizations was evaluated in terms of activity, isotacticity, H_2_ response, and comonomer incorporation, as will be discussed later. 

According to TGA analysis, the acquired MgCl_2_.nEtOH adducts contain different ethanol loadings of n = 0.7, 1.2, and 2.8, named Add-A, Add-B, and Add-C, respectively. These adducts were employed individually in the synthesis of ZN catalysts with similar reaction conditions. The synthesized catalysts were labeled as Cat-A, Cat-B, and Cat-C, consecutively. The XRD patterns of the prepared catalysts are shown in Figure 1. In the XRD pattern of Cat-A, several peaks are observed at 2θ angles of 15–16°, 20–45°, and 50°, which all represent the appropriate crystalline structure of the fabricated ZN catalyst [31], according to previous studies [32,33]. The peaks centered at 2θ angle of 15–16° demonstrate the presence of the (001) plane and correspond to the Cl–Mg–Cl triple layers along the crystallographic direction. The multiple sharp peaks observed at 2θ angles of 30–35° demonstrate the presence of (011), (012), and (104) planes and correspond to five-coordinated Mg atoms in α- and β-forms of MgCl_2_ [34,35,36]. Specifically, two peaks at 2θ = 32 and 34° are assigned to the (011) plane of β-MgCl_2_ (with hexagonal close packing, hcp) and (104) plane of α-MgCl_2_ (with cubic close packing, ccp), respectively. A comparison of the catalyst patterns shows that the peaks centered at 2θ angles of 30–35° are merged and transformed to a wide single peak by using an adduct containing a higher ethanol content (Cat-C, Figure 1). This result could be related to the vigorous reaction of titanium tetrachloride with ethanol, leading to the facile formation of the δ-form of MgCl_2_. It is well accepted that the δ-form is the most active in ZN catalysts [13,37,38].

In addition, the peak observed at the region of 2θ~50° demonstrates the presence of the (110) plane in the catalyst structure. The reduction in the intensity of this peak by moving from Cat-A to Cat-B and Cat-C reveals an increase in disorder in the structure of MgCl_2_, leading to easier titanation of the adduct. There is good agreement between XRD patterns of the current study and those obtained by Di Noto et al. [39], who explored structural features of different MgCl_2_(C_2_H_5_OH)_x_ adducts with 0 < x ≤ 6. 

The structural features of the as-synthesized ZN catalysts obtained from XRD patterns are summarized in Table 1. The d-spacing, full width at half-maximum (FWHM), crystallite size, and intensity of peaks at 2θ = 15, 32, and 50° are included in Table 1. These parameters are suitable indications of the MgCl_2_ crystalline size correlated with the (003), (011), and (110) planes, respectively. It is clear that for all of catalysts, the intensity of the peaks at 2θ = 32° are much higher than those at 2θ = 15 and 50°, which affirms a higher share of (011) surface compared to (003) and (110) planes. The crystallite size of the (011) plane (related to the peak at 2θ = 32°) diminished from 134 Å in Cat-A to 32 and 15 Å in Cat-B and Cat-C, respectively, containing a higher amount of ethanol in the primary adduct. This result could be attributed to an increase in disordered crystalline structures upon increasing the ethanol content of the adduct and subsequent thermal and chemical treatments. 

For the other two surfaces, i.e., (003) and (110), the crystallite size has a similar trend to that of the (011) plane and drops when the ethanol content of the adduct rises from 0.7 to 2.8 mol%. Additionally, the FWHM parameter decreases with increasing ethanol content of the adduct, indicating the enhancement of disorder in the crystallites of MgCl_2_. However, all three catalysts displayed almost similar d-spacing at each 2θ angle, though this feature was altered by varying the angle. 

The porosity of the as-synthesized catalysts was investigated through the use of N_2_ adsorption–desorption measurements. Figure 2 indicates a significant increase in the amount of adsorbed gas at lower pressures with an increasing molar ratio of ethanol in the primary adduct. Indeed, Cat-A had a lower hysteresis compared to Cat-B and Cat-C and revealed N_2_ adsorption isotherms of type III. This type of isotherm is indicative of meso-porous materials. In the related curves, the hysteresis loops are categorized into H_3_ type, which is suitable for conical-shape mesopores containing nonuniform shapes and sizes [40]. The isotherms of Cat-B and Cat-C are apparently classified as type IV. The hysteresis loop for these two samples belongs to the type H_2_, indicative of the presence of irregular mesopores in size and shape made by agglomeration of spheroidal particles [41]. Such type of isotherms is typical of industrial catalysts. The limited adsorption of N_2_ at higher relative pressures for such samples is an indication of filled pores. For the catalyst samples, the chemical reaction between Ti and alcohol of the adduct intensifies the de-alcoholation process, and thus, these samples displayed higher porosity. As indicated in Figure 2, the higher ethanol content of the adduct results in catalyst particles with higher porosity and vast surface area.

Table 2 summarizes the pore volume, size, and surface characteristics of the samples. Increasing the ethanol content of the MgCl_2_.nEtOH adduct leads to the enhancement of adsorbed gas volume and the total pore volume. In addition, the surface area of the catalysts increases markedly upon increasing the adducts ethanol content so that the surface area of Cat-C and Cat-B is seven and three times that of Cat-A, respectively. It should be noted that Cat-B and Cat-C samples exhibit a similar distribution of pores with an average diameter of 5 nm. However, the contribution of pores with a diameter ranging from 10 to 20 nm is larger for Cat-C than that of Cat-B. These results could be correlated to the vigorous reaction between TiCl_4_ and alcohol. The greater the ethanol content, the higher the reaction rate, leading to the increased number of large pores. Thus, the performance of the catalysts in propylene polymerization is expected to be markedly different. However, it was found that the average pore diameter of the catalyst samples declines from 21.51 to 4.32 nm when moving from Cat-A to Cat-C. This behavior could be attributed to a higher ratio of small to large pores for those samples, which was affirmed in the Barrett, Joyner, and Halenda (BJH) curve (Figure 2b).

The BJH curves in Figure 2b reveal significant modifications in the pore size depending on the ethanol content of the adduct. The pore size distribution (PSD) curves of the catalysts indicate a significant enhancement in the contribution of pores with lower diameters by increasing the ethanol content of the adduct. All catalyst samples exhibit a multimodal distribution of pores with a larger contribution of pores with diameters below 5 nm. Additionally, according to Figure 2b, the Cat-C sample shows a significantly higher number of large meso-pores at 10 < r_p_ < 22 nm that can directly influence the catalyst performance during copolymerization towards the comonomer response [42], which will be discussed later. Noteworthy, the catalyst samples display no pores with a diameter larger than 40 nm. This feature indicates good control in terms of the catalyst preparation process that has a critical role in the morphology and mechanical properties of the catalyst particles, resulting in polymeric particles with the desired morphology. 

Scanning Electron Microscopy (SEM) analysis was utilized to study the morphology of the prepared catalyst samples. As evidenced in the SEM pictures (Figure 3a,b), all catalysts had the spherical morphology that was expected for the catalyst particles synthesized from the MgCl_2_.nEtOH adducts [30]. The spherical catalyst particles consequently provide spherical polymer powders, advantageous to easy conveying in industrial reactors [36]. Moreover, the spherical morphology leads to increased bulk density and, thus, reducing the occupied volume. In addition, microparticles will not form and, therefore, dust explosions as well as reactor fouling will be minimized. On the other hand, all the catalysts display a cracked surface. This result could be correlated with the chemical treatment of the support particles with the TiCl_4_ precursor. Starting from the adduct containing a higher amount of ethanol, Add-C, provided a catalyst, Cat-C, which consists of an open bud-like disordered crystal structure displayed in Figure 3c. Moreover, Figure 3 demonstrates that the catalysts had not been broken, implying the precise and accurate procedure for the synthesis of the catalysts. The differences in porosity and three-dimensional structure of the catalysts will result in different catalytic activity as well as the comonomer content of the final polymers.

Table 3 shows the atomic composition of the catalysts obtained via elemental Energy-Dispersive X-Ray (EDX) analysis [43]. As expected from the previous results, the amount of Ti obtained for different samples directly corresponds to the ethanol present in the synthesized adducts. The Ti content of the catalysts is enhanced with the increasing ethanol dosage of the adduct. Actually, this result could be associated with the more vigorous de-alcoholation of Add-C and Add-B compared to Add-A, providing more free volume for the adsorption of the titanium precursor. Moreover, there is a considerable difference between the Mg and Cl content of the catalysts prepared with supports containing various initial amounts of ethanol. The Mg and Cl content in the percentage of the catalysts decreased from 11.30 to 7.99 and from 55.30 to 36.07 for Cat-A and Cat-C, respectively. This result could be attributed to the destruction of some parts of the support as the de-alcoholation reaction proceeds. In other words, de-alcoholation leads to the formation of planes of 4 and 5 coordinated Mg atoms that directly influence the amount of Cl. 

The C content of Cat-C was 39.63%, which is markedly higher than those of Cat-B and Cat-A, i.e., 27.70 and 20.24%, respectively. It should be noted that the C content is almost representative of the internal donor, i.e., DIBP and remained ethanol in the composition of the as-synthesized catalysts. The molar ratios of ethanol to MgCl_2_ support were 2.8, 1.2, and 0.7% for the Cat-C, Cat-B, and Cat-A samples, respectively. It is worth mentioning that in the catalyst synthesis step, ethanol reacts with TiCl_4_ and produces titanium halo ethoxide, which is soluble in TiCl_4_ and partially in hot toluene, so it is removed during catalyst synthesis. This reaction reduces the carbon content of the adduct, and the amount of reduction is directly dependent on the initial ethanol loading of the support. Therefore, the highest and lowest reduction in C content belong to Cat-C and Cat-A, respectively, which could originate from a more severe reaction between ethanol and TiCl_4_. However, the Cat-C sample showed the highest carbon content. 

Furthermore, the EDX data show that the oxygen loading of the catalyst increases from 9.44 to 12.17% with increasing the ethanol content of the adduct from 0.7 to 2.8 mol%. Oxygen could be considered as representative of the internal donor, i.e., DIBP. According to previous studies, the highly disordered MgCl_2_ structure tends more toward phthalate-based internal donors. As discussed before, the Cat-C sample displays the highest disorder in the series. Thus, this sample is expected to show the greatest adsorption of the electron donor. Moreover, the exchange reaction between ethanol and DIBP can favor the adsorption of oxygen on the catalyst. The substitution of butyl with ethyl group has been found to increase the adsorption of oxygen up to the time. Therefore, the Cat-C sample, which has the highest ethanol content, shows the highest oxygen content, which can greatly influence the stereoregularity of the final polymer that will be considered later. 

To investigate the distribution of the atoms on the catalyst surface, EDX mapping of samples was conducted. Figure 4 discloses that all atoms have been homogenously distributed on the surface of the Cat-A, Cat-B, and Cat-C samples.

The results of polymerization runs conducted on the as-synthesized catalysts are summarized in Table 4. Three polymerization batches were performed for each catalyst comprising (i) propylene monomer only, (ii) propylene monomer together with H_2_ as a chain transfer agent, and (iii) propylene and 1-hexene comonomer in the presence of H_2_. The data obtained via propylene homopolymerization reactions, represented by sample codes of P_A1_, P_B1_, and P_C1_, demonstrate that catalyst activity is highly dependent on the ethanol content and, consequently, on the surface area and pore size of the used adduct. In fact, the catalyst activity increases from 92.3 to 114.4 and then to 142.3 kg PP/(gTi·h) by increasing the ethanol loading of adduct from 0.7 to 1.2 and then to 2.8 mol%, respectively. This result could be associated with a higher surface area and Ti content of the catalyst with more ethanol loading, as supported by XRD and elemental analysis of the catalysts. In addition, according to the SEM photographs, the open bud morphology of the Cat-C sample facilitates the easier fragmentation of this catalyst, thus exposing more active sites to the monomer, resulting in higher catalyst activity.

Sozzanni et al. [44] also found similar results for the ethanol content of the catalyst. In addition, Sacchetti showed that in direct titanation, the ethanol/MgCl_2_ ratio of about 3 results in maximum catalyst activity [45].

Notably, the data in Table 4 show that polymerization in the presence of H_2_ as a chain transfer agent leads to the enhancement of the catalyst activity for all three catalysts, similar to the results reported in previous studies. This finding could be correlated to the hydrogenation of dormant species formed after 2,1-insertion of propylene monomer. It makes it possible for the so-called sites to reenter the catalytic cycle [46,47]. Interestingly, similar to the previous part, a direct relationship between catalyst activity and the adduct ethanol content can be seen in the presence of H_2_, too. Indeed, the catalyst productivity increases from 128.7 to 158.2 kg PP/(gTi.h) when moving from Cat-A to Cat-C. The enhancement of the catalyst activity upon the addition of H_2_ was less pronounced for the Cat-C sample. 

In the case of Cat-A and Cat-B samples, the copolymerization data of propylene and 1-hexene in the presence of H_2_, denoted by P_A3_ and P_B3_, respectively, showed a dramatic increase (1.7 and 1.3 times higher activity, respectively) in the catalyst activity from 92.3 to 159.5 and from 114.4 to 146.5 kg PP/(gTi.h), respectively. However, for the Cat-C sample, a milder increase (1.1 times) in catalyst activity from 142.3 to 161.1 kg PP/(gTi.h), was observed. The decrease in the growth of the catalyst activity in copolymerization experiments with the increase in the ethanol loading of adduct is in agreement with the previous data, i.e., homopolymerization in the presence of H_2_. The upward trend in catalyst activity in copolymerization runs could be related to (a) easier fragmentation of catalyst and (b) faster diffusion of monomer through the polymer to the active center of catalyst, originated by the decrease in polymer crystallinity upon the insertion of the comonomer in the polymer chain. 

Table 4 also summarizes the isotacticity indices of the synthesized polymers investigated using the Soxhlet technique, utilizing heptane as solvent. The isotacticity of the homopolymers synthesized in the absence of H_2_ using Cat-A, Cat-B, and Cat-C was found to be 88, 90, and 93%, respectively. These values are close to typical values for ZN catalysts containing DIBP as an internal donor. According to Table 4, although the isotacticity index of the homopolymers decreases in the presence of H_2_, it has an increasing trend with the ethanol content of the adduct. Despite the trend observed for the homopolymers, the copolymers showed a decrease in isotacticity upon increasing the adduct ethanol content. Furthermore, the copolymers exhibited lower isotacticity index values than both types of homopolymers. This result could be related to an increase in the amorphous content of the copolymers due to the comonomer presence. In addition, the higher comonomer content of the copolymers, in the presence of catalysts containing higher amounts of ethanol, contributed to the considerable drop in the isotacticity index of the copolymers when the ethanol loading of the adduct improved.

The increasing trend of the isotacticity of both series of homopolymers could be explained by the higher adsorption of donors on the surface of catalysts containing more ethanol [44,48,49]. In addition, the formation of side products during polymerization could result in an increase in the polymer isotacticity. For example, according to Harkonen et al. [50] and Lee et al. [15], alkoxysilane species formed by the interaction between the hydroxyl groups of the catalyst and the silane of the electron donor can participate in the polymerization as a second donor, enhancing the polymer isotacticity. Therefore, polymers with higher isotacticity are expected from catalysts containing larger amounts of ethanol.

Recently, Taniike et al. [40] have affirmed that in the presence of titanium halo alkoxide and ethanol compounds, the internal electron donor can participate in the trans-alkoxydification reaction so that the isobutyloxy group of the donor can be replaced by an ethoxy moiety of the mentioned precursors. To obtain more information about the structure of the catalysts, the ^1^H-NMR technique was utilized. Three spectra and peak assignments for three catalyst samples are shown in Figure 5. The peaks observed at chemical shifts of 0.9 and 1.9 ppm, indicated by numbers 1 and 2, respectively, represent protons from the butyl group of the esterified donor. The peaks at the chemical shifts of 4.2 and 7.6 ppm, denoted by numbers 3 and 4, respectively, represent protons from the ethyl group and the aromatic ring in the structure of the esterified donor.

The mole fraction of the butyl group in each catalyst, represented by n_Bu_, was calculated from the areas denoted by numbers 2 and 3 in Figure 5 and associated proton numbers, according to the following equations:nBu=Area22Area22+Area32

In the spectrum of the Cat-A sample (Figure 5c), no peak with a chemical shift of 4.2 ppm was observed, so the mole fraction of butyl was calculated based on the difference between the areas associated with the protons of the butyl group and the aromatic ring, i.e., areas denoted by the numbers 2 and 4, respectively. The area difference was found to be 0.076, which was attributed to the catalyst ethyl group that was barely detectable in the ^1^H-NMR spectrum. Therefore, the mole fraction of the butyl group in the catalyst sample Cat-A was calculated according to the following equation:nBu=Area22Area22+0.0762

The mole fractions of the butyl group in Cat-A, Cat-B, and Cat-C were found to be 86, 78, and 73%, respectively. The lower mole fraction of the butyl group in Cat-C implies that this catalyst contains more ethanol resulting from the higher ethanol content of Add-C. Therefore, more trans-alkoxydification reaction is expected in the reaction between the DIBP and ethanol of the MgCl_2_ support, resulting in the formation of a phthalate donor containing shorter hydrocarbons. As confirmed by Stukalov et al. [49], donors with shorter hydrocarbons are more likely to adsorb on the catalyst surface, yielding polymers with higher isotacticity.

In the following, to further identify the formation of bonds between the support and TiCl_4_ or internal donor, Fourier-transform infrared (FTIR) spectroscopy was considered. Figure 5d presents the FTIR spectrum of the Cat-A sample. In the related spectrum, the weak intensity signal at 2958 cm^−1^ is due to the stretching vibration of the C–H bonds. The strong signal at 1699 cm^−1^ is due to the C=O vibrations of the carbonyl group of the ester, revealing the deposition of the internal donor into the MgCl_2_ surface. The intense signal at 1616 cm^−1^ is due to the Mg-Cl stretching vibrations of the MgCl_2_ support. The signal at 1082 cm^−1^ is due to the asymmetric stretching vibrations of O-C=O. The weak signal at 615 cm^−1^ is attributed to the stretching vibrations of the Ti-Cl bond. The presented FTIR spectrum confirms the successful formation of bonds between the active catalyst and internal donor with MgCl_2_ surface. 

To further explore the role of ethanol, Density Functional Theory (DFT) simulations have been carried out to compare the binding energy and catalytic activity of DIBP and DEP towards propylene insertion. 

The surface models used in this work have previously been employed in different studies evaluating the effect of aluminum co-catalysts [43,51]. Other surface models previously evaluated [52] were the co-catalyst is placed on Ti cluster chlorine atoms, with the goal to induce the metal–metal bond between Ti and Al. However, our results proved that this particular adsorption site is higher in energy (16 kcal/mol) than the direct adsorption of the co-catalyst on the MgCl_2_ surface model. This is because the number of contacts between the co-catalyst molecule with Mg, and chlorine atoms stabilize the system. To determine the binding energy of DIBP and DEP on our model of the MgCl_2_ surface, we have performed an exhaustive conformational search. The most stable configurations were tested to find the most energetically favored orientation of DIBP and DEP on the MgCl_2_ surface. Indeed, it is important to mention that convergence has only been reached for structures that have both O atoms of the open 6-membered ring of DIBP and DEP pointing to the surface, anchored on Mg atoms, emphasizing that the surface adsorbate interactions favor the co-catalysts adsorption, instead of Ti-DIBP and Ti-DEP interactions. This adsorption orientation implies that the phenyl group is then placed parallel to the surface, and the ether group is placed further away from the MgCl_2_. The binding energy values are −45.8 and −43.6 kcal/mol for DIBP and DEP, respectively, revealing a strong adsorbate–surface interaction for both compounds. No remarkable differences have been observed in the adsorption geometry, with the Mg-O distance ranging between 2.03 and 2.10 Ǻ, and thus independent of the electron donor adduct. 

The reaction mechanism of propylene polymerization is performed using the common TiCl_4_ cluster once activated with the ligand -CH_2_CH(CH_3_)_2_ on the titanium, supported on MgCl_2_ surface model (**I**), and thus the fifth ligand is an isobutyl moiety bonded to the Ti metal center. The propylene coordination on the Ti cluster (**II**) is evaluated as the initial step before the insertion into the olefin to generate the carbon chain (**III**). The kinetic energy barrier from **II** to **III** has been calculated by means of the search for a transition state structure, which contains only one imaginary frequency corresponding to the C–C bond formation. The catalytic pathway has been computed twice, considering the *si* and *re* transition state forms of the propylene on the reaction energetics since it can be directly related to the polypropylene tacticity. Consequently, there is the trend that the bigger the E_TS-*re*_-E_TS-*si*_ difference, the higher the polypropylene tacticity. To evaluate the effect of DIBP and DEP, a model that contains two molecules of both compounds has been anchored independently on the right and left side of the TiCl_4_ cluster, as illustrated in Figure 6.

Focusing on the effect of DIBP, Figure 7 shows the importance of the *si* and *re* forms on the energetics of the transition state structure. Both forms have a favorable interaction energy with the supported TiCl_4_(CH_2_CH(CH_3_)_2_) cluster, with these energies being −3.55 and −3.38 kcal/mol for *re* and *si* forms, respectively. A difference of 0.92 kcal/mol was obtained for the transition state structures of *si* and *re* forms. On the other hand, a different tendency was found for the DEP electron donor. First of all, the interaction of *si* and *re* forms with the TiCl_4_(CH_2_CH(CH_3_)_2_) supported on the MgCl_2_ cluster shows a slightly favored interaction in terms of *re* orientation with respect to the adduct in the *si* form. It can be observed that both transition state structures have relative Gibbs energies larger than the reference in comparison to the model that uses DIBP as an electron donor. In addition, the energy difference of both transition state structures is only 0.30 kcal/mol, showing a lower difference with respect to the use of DIBP. Therefore, the bigger E*_re_*_-*si*_ difference is found for DIBP, predicting higher PP tacticity with respect to the use of DEP as an electron donor, which is in agreement with experimental results. In order to investigate whether there was any possibility of understanding the results and the minimal difference between DEP and DIBP, a series of analyses were undertaken. Structurally, the %V_Bur_ parameter and the steric maps of Cavallo and coworkers [53] (see Appendix A) denote that there is no significant steric difference between DEP and DIBP. Electronically, through Natural Population Analysis (NPA) charges, we found no difference between the two comonomer typologies. It should be borne in mind that such subtle energy differences are almost or even exceeding the DFT limits of accuracy.

The melting point, crystallization temperature, and crystallinity of the copolymers, measured using DSC, are also summarized in Table 5, and the corresponding diagrams are shown in Figure 8.

According to Table 5, copolymers synthesized from Cat-A and Cat-B, that is, P_A3_ and P_B3_, had similar melting points and crystallization temperatures of ~153 °C and ~108 °C, respectively. While the copolymer sample P_C3_, synthesized utilizing Cat-C, showed slightly lower melting and crystallization points of ~150 °C and 104 °C, respectively. This is in agreement with previous studies that confirmed that both isotacticity and comonomer content influence the melting point, crystallization temperature, and crystallinity of a copolymer [54].

The lower crystallinity of sample P_C3_ could be related to its lower isotacticity compared to the other two samples [55,56]. In addition, a related catalyst had an open-bud morphology, a greater content of large pores as well as a markedly greater surface area, which all enable the easier incorporation of 1-hexene comonomer to the growing chain and thereby a reduction in copolymer crystallinity [40]. The so-called Filter effect can also affect the comonomer content of the copolymers [57]. This can suppress the comonomer content of the P_A3_ and P_B3_ samples that have smaller pores compared to sample P_C3_. 

To evaluate the impact of the adduct ethanol content on the sequence length distribution of the synthesized PPs and to reach a deeper understanding of the polypropylene chain microstructure [58,59,60], Singular Spectrum Analysis (SSA) was performed. Figure 9 shows the melting diagrams of the copolymer samples synthesized in the presence of H_2_, i.e., P_A3_, P_B3_, and P_C3_, after SSA treatment, as well as the de-convoluted curve for P_B3_. The melting points (T_m_) related to each peak are listed in Table 6. As can be seen in the SSA profiles, multiple melting peaks for the synthesized PPs related to the melting of crystallites with different lamellar thickness, L, and containing chains with different meso (or isotactic) sequence lengths, MSL. The relative content of each T_m_ peak (n%, obtained by the Origin2020 software), together with its related MSL and L, calculated using appropriate formulas indicated in the previous literature [61] are also given in Table 6.

The SSA curves exhibit four melting peaks for the copolymer samples P_A3_ and P_B3_, while only three peaks were observed for the copolymer sample P_C3_. Additionally, the lamellar thickness of 10.61, 10.14, and 7.98 nm for P_A3_, P_B3_, and P_C3_, respectively, implies that P_C3_ has the greatest comonomer content. Moreover, based on the values obtained for n, the principal melting peak for P_A3_ and P_B3_, is around 157 °C, while it is around 139 °C for P_C3_. This result, in agreement with previous data [18], could be related to the higher comonomer content of P_C3_, leading to the lowest MSL values for this sample, according to Table 6. Indeed, the chains with lower MSL provide crystallites with higher amounts of defects, resulting in thinner lamellar thicknesses and lower melting temperatures. Furthermore, for P_C3_, the melting peaks had the same height, corresponding to similar n values, as shown in Table 6, indicating a more uniform distribution of the comonomer in the structure of P_C3_ compared to P_A3_ and P_B3_.

These findings demonstrate that the ethanol content of the primary adduct has a strong influence on the comonomer incorporation of catalyst and thus on the microstructure of the obtained copolymers.

Furthermore, it was revealed that by increasing the adducts ethanol loading, melting enthalpy (ΔH_m_) of the copolymers diminishes from 88.15 to 58.04 J/g. This could originate from the higher comonomer content of the samples synthesized using the Cat-C catalyst, resulting in the increase in the copolymers’ amorphous region. This outcome was correlated with the higher ratio of large pores in Cat-C compared to Cat-B and Cat-A, which facilitates diffusion of large 1-hexene comonomer to reach the active growing chain. 

Then, to analyze in depth the data of the SSA patterns, the statistical parameters of the synthesized copolymers [62,63], including the number and the weighted averages of the isotactic sequence length (MSL_n_ and MSL_w_, respectively), the number and weighted averages of lamellar thickness, i.e., L_n_, and L_w_, as well as the amplitude index (I), were calculated and summarized in Table 6. In agreement with previous results, increasing the adducts ethanol content leads to copolymers with lower L_n_, L_w_, MSL_n_, and MSL_w_ amounts with a more unimodal distribution. This result could be correlated with a more uniform distribution of the comonomer within copolymer chains synthesized from catalysts that have higher amounts of ethanol. Such uniform comonomer distribution for P_C3_, as shown by MSL_w_/MSL_n_ values, furnishes higher mechanical properties.

To evaluate the influence of the catalyst ethanol content on the chemical composition distribution of the copolymers, Temperature Rising Elution Fractionation (TREF) analysis was employed, and the results are illustrated in Figure 10. Using this method, the polymer chains are fractionated according to their crystallinity differences caused by their varied comonomer content as well as various regio and stereo defects. Based on the TREF profiles shown in Figure 10, the copolymer from the catalyst with the lowest ethanol loading, i.e., P_A3_, displayed a peak value at the elution temperature above 100 °C, an indication of a PP homopolymer [64,65]. However, at low elution temperatures of <100 °C, copolymer P_C3_ shows the highest fractionation levels of about 55.1%. This result is in line with the lowest MSL and lamellar thickness of this sample compared to P_A3_ and P_B3_, caused by its higher comonomer content and lower stereo regularity, as demonstrated in SSA analysis. The results of the measurement of the crystallinity of the samples are in good agreement with these findings, supporting them in a promising way. 

## 7. Conclusions

This work provided a deep examination of how the ethanol content of MgCl_2_.nEtOH adducts within Ziegler–Natta (ZN) catalysts influence their performance in propylene homo- and copolymerizations. The investigation encompasses activity levels, isotacticity, response to hydrogen (H_2_), and the incorporation of comonomer, i.e., 1-hexene. Three different MgCl_2_.nEtOH adducts were synthesized, possessing n values of 0.7, 1.2, and 2.8. These adducts were subsequently employed in the creation of respective ZN catalysts.

The characterization of the catalysts was conducted using a broad range of experimental techniques, unveiling the microstructure of the synthesized (co)polymers, as well as DFT methods that rationalized the kinetics and thermodynamics of the polymerization. The outcomes revealed that activity levels increased as the ethanol content of the adduct rose, both in homo- and copolymerization reactions. This increase was particularly significant in homopolymerization reactions conducted without the presence of H_2_.

Additionally, it was observed that the catalyst with the highest ethanol content produced a copolymer featuring a lower isotacticity index, shorter meso sequence length, and a more even distribution of comonomer within the polymer chains. These effects were attributed to several factors, including the greater total surface area and titanium content of the corresponding catalyst. Moreover, the catalyst’s lower average pore diameter, greater proportion of large pores relative to the other catalysts, and its distinctive spherical open bud morphology played key roles in these results.

In conclusion, this research underscores the significance of managing the ethanol content of catalyst/support during the preparation process, as it significantly impacts the catalyst’s performance in terms of both activity and product properties in propylene homo- and copolymerizations. The findings indicate that higher ethanol content correlates with an increased loading of Ti catalyst onto the magnesium chloride support, resulting in enhanced catalytic activity. Nevertheless, the elevated ethanol content also results in reduced stereochemical control.

## Figures and Tables

**Figure 1 polymers-15-04476-f001:**
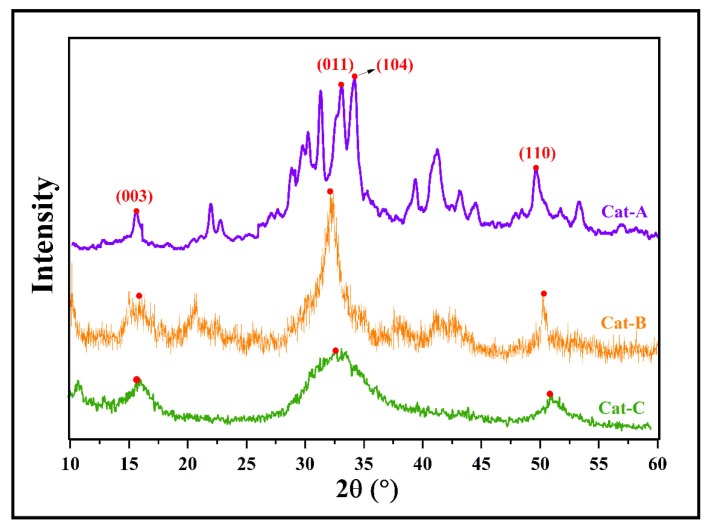
XRD patterns of the prepared catalysts utilizing adducts with different ethanol content (the red dots refer to (003), (011), (104) and (110) surfaces).

**Figure 2 polymers-15-04476-f002:**
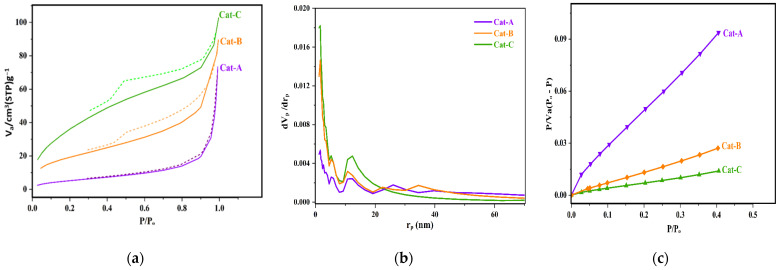
(**a**) N_2_ adsorption–desorption isotherms, (**b**) pore size distribution profile, and (**c**) standard diagram of the studied catalysts.

**Figure 3 polymers-15-04476-f003:**
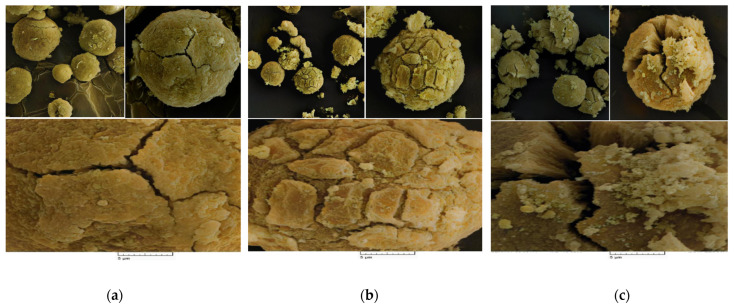
SEM photographs of the (**a**) Cat-A, (**b**) Cat-B and (**c**) Cat-C samples (SEM HV = 20.0 kV, SEM MAG = 10.00 kx, WD = 17 mm).

**Figure 4 polymers-15-04476-f004:**
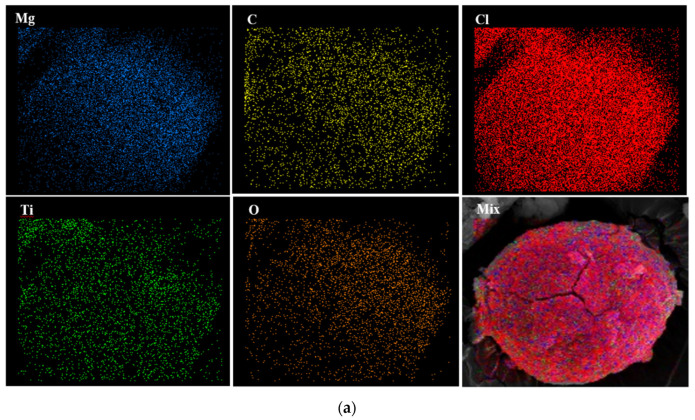
EDX elemental distribution of (**a**) Cat-A, (**b**) Cat-B and (**c**) Cat-C samples.

**Figure 5 polymers-15-04476-f005:**
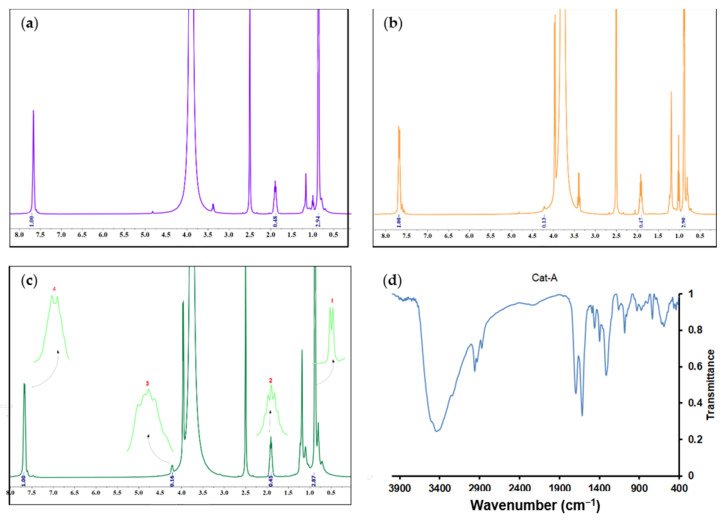
^1^H-NMR spectra (in ppm) of the synthesized catalysts: (**a**) Cat-A, (**b**) Cat-B, and (**c**) Cat-C, obtained in DMSO-d6 solvent. (**d**) FTIR spectrum of Cat-A sample.

**Figure 6 polymers-15-04476-f006:**
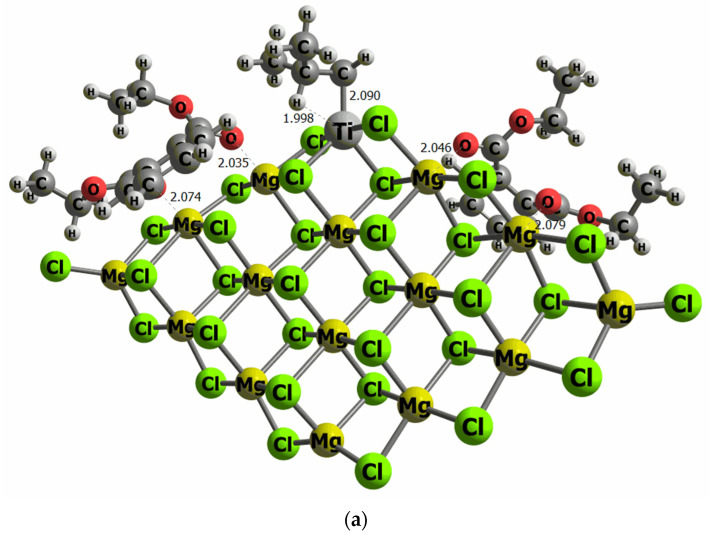
Computational models of the TiCl_4_ supported on the MgCl_2_ surface containing two molecules of DEP (**a**) and DIBP (**b**) adsorbed at both sides of the TiCl_4_ cluster (selected distances in Ǻ).

**Figure 7 polymers-15-04476-f007:**
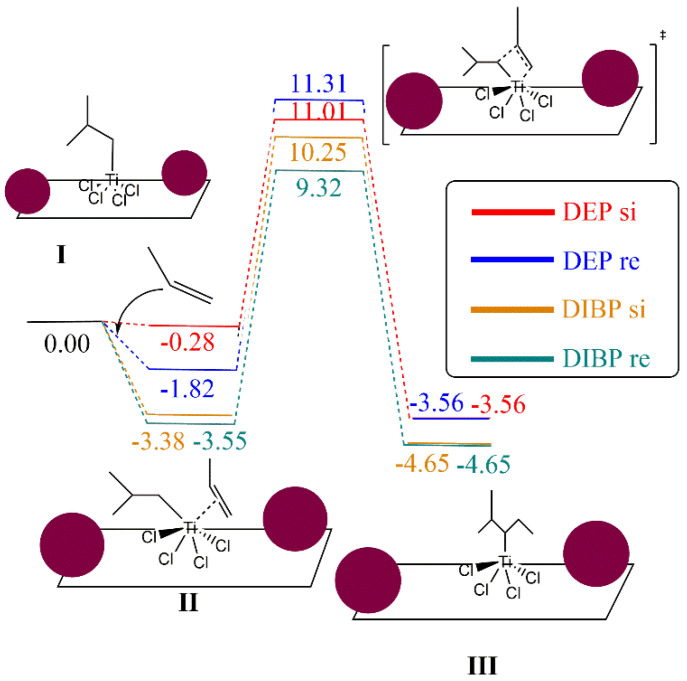
Gibbs energy profile of propylene polymerization using TiCl_4_ supported on MgCl_2_ surface. Energies are in kcal/mol. Maroon balls represent the electron donor structures DEP and DIBP.

**Figure 8 polymers-15-04476-f008:**
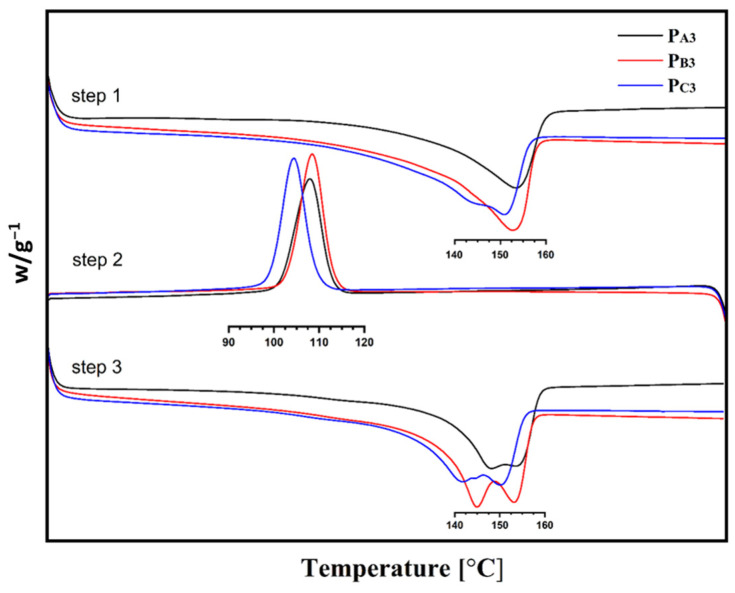
DSC curves of the synthesized PP samples.

**Figure 9 polymers-15-04476-f009:**
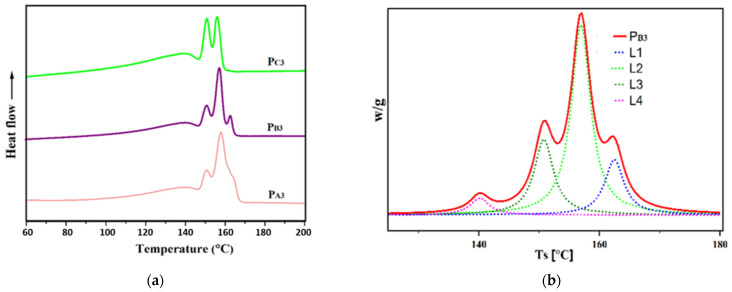
(**a**) SSA curves of the propylene/1-hexene copolymers obtained from synthesized catalysts and (**b**) de-convoluted SSA curves for copolymer synthesized from Cat-B.

**Figure 10 polymers-15-04476-f010:**
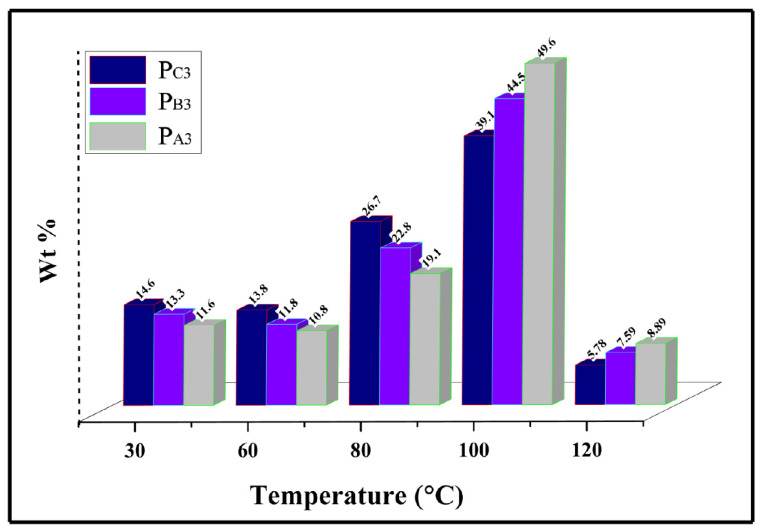
TREF analysis of the synthesized copolymers.

**Table 1 polymers-15-04476-t001:** Structural features of the synthesized Ziegler–Natta catalysts obtained from XRD patterns.

Catalyst Type	Characteristic	2θ Angle [°] (Related Plane)				
		50 (110)	34 (104)	32 (011)	30 (012)	15 (003)
Cat-A	d-Spacing [Å]	1.84	2.63	2.72	2.86	5.69
	FWHM [°]	0.47	0.55	0.62	0.29	0.62
	Crystallite size [Å]	186	151	134	284	129
	Intensity [%]	201	378	367	352	132
Cat-B	d-Spacing [Å]	1.80	-	2.77	-	5.51
	FWHM [°]	1.58	-	2.50	-	0.72
	Crystallite size [Å]	56	-	32	-	111
	Intensity [%]	109	-	351	-	99
Cat-C	d-Spacing [Å]	1.77	-	2.67	-	5.40
	FWHM [°]	1.88	-	5.70	-	1.06
	Crystallite size [Å]	47	-	15	-	76
	Intensity [%]	96	-	287	-	87

**Table 2 polymers-15-04476-t002:** Porosity characteristics of the as-synthesized catalyst samples.

Sample	Surface Area (m^2^/g)	Total Pore Volume (cm^3^/g)	Average Pore Diameter (nm)	Adsorbed Gas Volume (cm^3^/g)
Cat-A	20.21	0.108	21.51	4.64
Cat-B	69.5	0.131	7.56	15.9
Cat-C	141.6	0.153	4.32	32.5

**Table 3 polymers-15-04476-t003:** Elemental analysis results of the synthesized catalysts obtained from EDX.

Sample	Ti (%)	Mg (%)	Cl (%)	O (%)	C (%)
Cat-A	3.63	11.30	55.30	9.44	20.24
Cat-B	3.89	9.39	48.20	10.78	27.70
Cat-C	4.14	7.99	36.07	12.17	39.63

**Table 4 polymers-15-04476-t004:** Characterization of the obtained polymers via synthesized catalysts ^1^.

Polymer Label	Catalyst Type	1-Hexene (mL)	H_2_ (bar)	Activity kg PP/(g Ti.h)	II ^2^	M_n_ (g/mol)	Ð
P_A1_	Cat-A	-	-	92.3	88	30,500	4.7
P_A2_	Cat-A	-	3	128.7	82	20,800	4.8
P_A3_	Cat-A	5	3	159.5	78	21,100	5.1
P_B1_	Cat-B	-	-	114.4	90	28,900	5.2
P_B2_	Cat-B	-	3	126.2	85	18,500	4.8
P_B3_	Cat-B	5	3	146.5	74	19,200	5.1
P_C1_	Cat-C	-	-	142.3	93	26,700	5.7
P_C2_	Cat-C	-	3	158.2	89	17,400	5.2
P_C3_	Cat-C	5	3	161.1	65	17,200	5.6

^1^ Polymerization condition: Al/Ti = 180 mol/mol, prepolymerization time = 10 min, prepolymerization pressure = 2 bar, prepolymerization temperature = 35 °C, TEAL/C-Donor = 20 mol/mol, polymerization temperature = 70 °C, polymerization pressure = 6 bar, 1-hexene = 5 mL, polymerization time = 50 min. ^2^ Isotacticity index, heptane soluble fraction.

**Table 5 polymers-15-04476-t005:** Melting temperature (T_m_), lamellar thickness (L_c_), meso sequence length (MSL) and relative content of each peak (n) obtained from SSA fractionated samples.

	Peak NO	T_m_ (°C)	l_c_ (nm)	MSL	n (%)	ΔHm (J/g)
P_A3_	1	163.4	10.61	49.00	12.35	88.15
	2	157.9	8.58	39.60	40.07	
	3	151.1	6.91	31.93	12.00	
	4	139.3	5.21	24.07	35.56	
P_B3_	1	162.3	10.14	46.80	7.91	93.72
	2	156.9	8.28	38.23	39.25	
	3	150.8	6.89	31.82	16.07	
	4	139.1	5.20	24.00	36.75	
P_C3_	1	155.7	7.98	36.83	31.74	58.04
	2	150.7	6.87	31.73	28.48	
	3	139.7	5.26	24.27	39.76	

**Table 6 polymers-15-04476-t006:** Statistical parameters related to lamellar thickness and MSL of PP copolymers.

	Ln (nm)	Lw (nm)	Lw/Ln	MSLn	MSLw	MSLw/MSLn
P_A3_	6.50	6.95	1.068	34.3	36.5	1.06
P_B3_	6.35	6.70	1.054	32.6	34.3	1.05
P_C3_	6.17	6.37	1.019	30.3	31.3	1.03

## Data Availability

Details of the experimental characterization via SSA and HNMR analyses. The computational data that support the findings of this study are available in the Appendix A of this article, including the xyz coordinates and absolute energies of all computed species.

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
