# Peer review of "Influence of the Ethanol Content of Adduct on the Comonomer Incorporation of Related Ziegler–Natta Catalysts in Propylene (Co)polymerizations"

_polymers, 2023, doi:10.3390/polym15234476_

Round 1
Reviewer 1 Report
Comments and Suggestions for Authors
Heterogeneous group IV transition metal catalyst represented by Ziegler-Natta (ZN) catalyst has been one of the main commercial catalysts for producing HDPE, LLDPE and iPP since 1950s. The third/fourth generation ZN catalysts based on magnesium chloride support are mainly used at present. In this manuscript, the author studied the influence of ethanol content in the preparation process of magnesium chloride support on catalyst loading and catalytic performance in propylene polymerization and copolymerization with 1-hexene. The results show that the increase of ethanol content can increase the loading of Ti catalyst on magnesium chloride support, which leads to the improvement of catalytic activity. However, the increase of ethanol content also leads to the decline of stereochemical control. Before publication, the authors should address the following issues:
1. Although polyolefins prepared by ZN catalyst are widely used, the authors should specify the use of propylene/1-hexene copolymer, a less common product.
2. The stereochemical control of propylene polymerization in the manuscript is significantly lower than the technical level of the current commercial ZN process. In fact, this is even lower than the technical level many years ago.
3. From the perspective of chemistry, the comparison of catalytic activity is incomplete. The authors should also make a comparison based on the unit of Kg PP/(g Tiâ–ªh) or Kg PP/(mol Tiâ–ªh).
4. For propylene/1-hexene copolymerization, the chemical composition (molar ratio of 1-hexene units) of the copolymers should be provided.
5. In Figure 5, are they liquid or solid NMR spectra of catalysts? If they are the former, the results are meaningless.
6. The cluster model used in DFT calculation should be supported by literatures. In addition, does the author consider the influence of solvent in DFT calculation of single-point energy?
7. The charge obtained based on NBO analysis should be called NPA (natural population analysis) charge.
Author Response
Comments and Suggestions for Authors
Heterogeneous group IV transition metal catalyst represented by Ziegler-Natta (ZN) catalyst has been one of the main commercial catalysts for producing HDPE, LLDPE and iPP since 1950s. The third/fourth generation ZN catalysts based on magnesium chloride support are mainly used at present. In this manuscript, the author studied the influence of ethanol content in the preparation process of magnesium chloride support on catalyst loading and catalytic performance in propylene polymerization and copolymerization with 1-hexene. The results show that the increase of ethanol content can increase the loading of Ti catalyst on magnesium chloride support, which leads to the improvement of catalytic activity. However, the increase of ethanol content also leads to the decline of stereochemical control. Before publication, the authors should address the following issues:
OUR ANSWER: It should be noted that the summary is very insightful and in fact, we have used an excerpt from it to clarify the message of ethanol insertion: “The findings indicate that higher ethanol content correlates with an increased loading of Ti catalyst onto the magnesium chloride support, resulting in enhanced catalytic activity. Nevertheless, the elevated ethanol content also results in reduced stereochemical control.”
- Although polyolefins prepared by ZN catalyst are widely used, the authors should specify the use of propylene/1-hexene copolymer, a less common product.
OUR ANSWER: We agree that propylene/ethylene is among the most used combination, but for industry the combination with 1-hexene is especially remarkable. Actually, alkyl chain of long α-olefin comonomer improves the final polymer’s mechanical properties. On the other hand, in ZN assisted olefin polymerizations, α-olefin incorporation is a major challenge, see our recent article at “J. Ind. Eng. Chem. 2022, 116, 359-370”, which suppresses these catalysts performances. Improving the related catalyst’s features to successfully incorporate long α-olefins was always followed by the researchers. In this regard, we have focused on ethanol content of adduct to see its effect on the incorporation of 1-hexene, as long alkyl side chain containing comonomer.
- The stereochemical control of propylene polymerization in the manuscript is significantly lower than the technical level of the current commercial ZN process. In fact, this is even lower than the technical level many years ago.
OUR ANSWER: We agree with this, and consequently we have included this in the conclusion: “The findings indicate that higher ethanol content correlates with an increased loading of Ti catalyst onto the magnesium chloride support, resulting in enhanced catalytic activity. Nevertheless, the elevated ethanol content also results in reduced stereochemical control.”
- From the perspective of chemistry, the comparison of catalytic activity is incomplete. The authors should also make a comparison based on the unit of Kg PP/(g Tiâ–ªh) or Kg PP/(mol Tiâ–ªh).
OUR ANSWER: Following the reviewer suggestion, the activities were expressed by kg PP/(g Tiâ–ªh) units in Table 4.
- For propylene/1-hexene copolymerization, the chemical composition (molar ratio of 1-hexene units) of the copolymers should be provided.
OUR ANSWER: Actually, due to the limited availability to the hot CNMR instrument, necessary for polyolefin materials due to their low solubility, we emphasized onto the melting temperature (Tm), lamellar thickness (Lc), and meso sequence length (MSL) data, obtained via SSA analysis. The data were gathered in Table 5.
- In Figure 5, are they liquid or solid NMR spectra of catalysts? If they are the former, the results are meaningless.
OUR ANSWER: It is solution HNMR analysis. The procedure is a well-known method, suggested by Taniike et al. [J. Catal. 2014, 311, 33-40] in order to identify in which extent the internal donor reacts with the ethanol of the adducts through trans-alkoxidification reaction. As a result of such reaction, isobutyloxy group of the donor can be replaced by an ethoxy moiety of the ethanol. The results show good agreement with the ethanol content, so that, higher ethanol content enables higher extent of so-called reaction.
- The cluster model used in DFT calculation should be supported by literatures. In addition, does the author consider the influence of solvent in DFT calculation of single-point energy?
OUR ANSWER: We have added a new paragraph, including new references 51 and 52 to explain the cluster model and that also show that those calculations rule out the solvent effect since the gas-phase approach is closer to the reality in the surface: “The surface models used in this work have previously been employed in different studies evaluating the effect of aluminum co-catalysts [43,51]. Other surface models previously evaluated [52] were the co-catalyst of Ti cluster chlorine atoms, emphasizing the metal-metal bond between Ti and Al. However, our results proved that this particular adsorption site is higher in energy than the direct adsorption of the co-catalyst on the MgCl2 surface model. This is because the number of contacts between the co-catalysts molecule with Mg and chlorine atoms stabilize the system. To determine the binding energy of DIBP and DEP on our model of MgCl2 surface, we have performed an exhaustive conformational search. The most stable configurations were tested to find the most energetically favored orientation of DIBP and DEP on the MgCl2 surface. Indeed, it is important to mention that convergence has only been reached for structures that have both O atoms of the open 6-membered ring of DIBP and DEP pointing to the surface, surface-anchored Mg atoms, emphasizing than the surface adsorbate interactions favor the co-catalysts adsorption, instead of Ti-DIBP and Ti-DEP interactions.”
- The charge obtained based on NBO analysis should be called NPA (natural population analysis) charge.
OUR ANSWER: We agree with the referee and we apologize for this error, and we change the text accordingly “Natural Population Analysis (NPA) charges”.
Reviewer 2 Report
Comments and Suggestions for Authors
Posada-Perez and coworkers have conducted a detailed study on the catalyst structure and its effect on Ziegler Natta (co)polymerizations of propylene and 1-hexene. The MgCl2.xEtOH/TiCl4 catalysts have been extensively characterized by XRD, EDX Mapping, SEM and DFT. The structure-property relationship between the ZN catalysts and the resulting polypropylenes have been highlighted. This study is suitable for publication in Polymers.
Comments/Suggestions:
1. Page 2 Line 65: Are there any other studies on varying the alcohol content ? If so please cite, if not, then add a statement.
2. Page 10 Line 341: Please specify what solvent was used for the NMR experiment.
Author Response
Comments and Suggestions for Authors
Posada-Perez and coworkers have conducted a detailed study on the catalyst structure and its effect on Ziegler Natta (co)polymerizations of propylene and 1-hexene. The MgCl2.xEtOH/TiCl4 catalysts have been extensively characterized by XRD, EDX Mapping, SEM and DFT. The structure-property relationship between the ZN catalysts and the resulting polypropylenes have been highlighted. This study is suitable for publication in Polymers.
OUR ANSWER: We thank the positive comments.
Comments/Suggestions:
- Page 2 Line 65: Are there any other studies on varying the alcohol content ? If so please cite, if not, then add a statement.
OUR ANSWER: Apart from the reference included in the manuscript we have added a new paper that some of us we submitted for consideration as new reference 18.
- Page 10 Line 341: Please specify what solvent was used for the NMR experiment.
OUR ANSWER: The solvent type for HNMR solvent is provided at Figure 5 caption.
Reviewer 3 Report
Comments and Suggestions for Authors
The manuscript submitted intends to help understand the effect of ethanol in the ZN catalysts. The paper is well-written with goals defined in the Introduction. The methods used to conduct the research were appropriated. I have some questions to improve the quality of the paper:
line 88- please, consider change the expression "' mentioned by us" to "as already reported".
Table 1- it is not possile to compara intensity among catalysts because the total value for each catalyst is different. Please, consider normalize these data.
- Please, describe in the methods section how ethanol content was determed.
- I kindly suggest to include FTIR and/or Raman to identify the possible formation of Ti-OR and/or Ti-O-Ti moieties in the catalysts.
- Text in line 252-256 is not clear. Is carbon content related to donor only?
- How donor content was determined in the catalysts?
- Please, include activity as kgPP/molTi/h? It is better to understand about the quality of the acive site from this data rather than kgPP/gcat/h.
- Please, discuss how the copolymerization activity is related to the average pore size of the catalysts?
- Please, include in the methods section how H-NMR and DSC/SSA were determined.
I kindly suggest the acceptation of this paper after the review from authors.
Comments on the Quality of English LanguageThe English is OK.
Author Response
Comments and Suggestions for Authors
The manuscript submitted intends to help understand the effect of ethanol in the ZN catalysts. The paper is well-written with goals defined in the Introduction. The methods used to conduct the research were appropriated. I have some questions to improve the quality of the paper:
line 88- please, consider change the expression "' mentioned by us" to "as already reported".
OUR ANSWER: We thanks all the comments that have improved the manuscript, once solved the errors.
Table 1- it is not possile to compara intensity among catalysts because the total value for each catalyst is different. Please, consider normalize these data.
OUR ANSWER: We apologize with the reviewer for this inconsistency. However, since we need the neat values, data normalization is not possible.
- Please, describe in the methods section how ethanol content was determined.
OUR ANSWER: Following the reviewer suggestion, the employed procedure for the determination of ethanol content of the adducts was added to the “Characterization” section in the SI file:
Thermogravimetric analysis (TGA) was employed to explore thermal stability of the adduct samples in the range of 25–400 °C by utilizing TGA instrument (Mettler Toledo Inc., Switzerland) under N2 atmosphere and with the heating rate of 10°C/min. The adduct samples showed two step weight loss in the range of 150-250 °C, which is attributed to the EtOH desorbtion from the adduct samples. [S. Patthamasang, B. Jongsomjit, P. Praserthdam. Efect of EtOH/MgCl2 molar ratios on the catalytic properties of MgCl2-SiO2/TiCl4 Ziegler–Natta catalyst for ethylene polymerization. Molecules 2011, 16, 8332–8342]. By calculating weight loss between the aforementioned temperatures, the ethanol content was determined.
- I kindly suggest to include FTIR and/or Raman to identify the possible formation of Ti-OR and/or Ti-O-Ti moieties in the catalysts.
OUR ANSWER: Following reviewer comment, the FTIR spectrum of Cat-A was added to the revised manuscript. The related explanation was included in Page 12 as the following.
To identify the formation of bonds between the support and TiCl4 or internal donor, FTIR spectroscopy was considered. Figure 5d presents the FTIR spectrum of the Cat-A sample. In the related spectrum the weak intensity signal at 2958 cm-1 is due to the stretching vibration of the C–H bonds. The strong signal at 1699 cm-1 is due to the C=O vibrations of the carbonyl group of the ester, revealing the deposition of internal donor into the MgCl2 surface. The intense signal at 1616 cm-1 is due to the Mg-Cl stretching vibrations of the MgCl2 support. The signal at 1082 cm−1 is due to the asymmetric stretching vibrations of O-C=O. The weak signal at 615 cm−1 is attributed to the stretching vibrations of Ti-Cl bond. The presented FTIR spectrum confirms the successfull formation of bonds between active catalyst and internal donor with MgCl2 surface.
Figure 5. d) FTIR spectrum of the Cat-A sample.
- Text in line 252-256 is not clear. Is carbon content related to donor only? How donor content was determined in the catalysts?
OUR ANSWER: We rephrased a bit those sentences. Actually, the reviewer is right and we did not determine the exact donor content in the synthesized catalysts. Rather than that, the amount of butyl to ethyl group in in the internal donor composition was determined quantitatively by HNMR spectroscopy and reported in the text related to Figure 5 explanation. It is worth mentioning that the calculated C content, obtained via elemental analysis, is almost representative of both the internal donor, i.e. diisobutylphtalate (DIBP), and remained ethanol in the composition of the as synthesized catalysts [Microchim. Acta 1988, 95, 85–87].
- Please, include activity as kgPP/molTi/h? It is better to understand about the quality of the acive site from this data rather than kgPP/gcat/h.
OUR ANSWER: According to the reviewer comment, activity values were represented by kg PP/(g Ti.h) unit. Also, the values in the corresponding text were edited accordingly.
- Please, discuss how the copolymerization activity is related to the average pore size of the catalysts?
OUR ANSWER: Explanation correlating the polymerization activity to the ethanol content, surface area and pore size is included in Page 9 of the revised manuscript.
The data obtained via propylene homopolymerization reactions, represented by sample codes of PA1, PB1, and PC1, demonstrate that the catalyst activity is highly dependent on the ethanol content, and consequently on the surface area and pore size of the used adduct. In fact the catalyst activity increases from 92.3 to 114.4 and then 142.3 kg PP/(gTi.h), increasing the ethanol loading of adduct from 0.7 to 1.2 and then 2.8 mol%, respectively. This result could be associated to a higher surface area and Ti content of the catalyst with more ethanol loading as supported by XRD and elemental analysis of the catalysts.
- Please, include in the methods section how H-NMR and DSC/SSA were determined.
OUR ANSWER: The procedures for SSA and HNMR analyses had already been provided in the Supporting information file, “Characterization” section. But since it was unclear, we added a comment in the manuscript, in particular in the Data Availability Statement.
I kindly suggest the acceptation of this paper after the review from authors.
OUR ANSWER: We thank again the comments and the advice to improve the manuscript.
Comments on the Quality of English Language
The English is OK.
OUR ANSWER: We thank the positive evaluation.
Reviewer 4 Report
Comments and Suggestions for Authors
1 A brief summary:
This article is devoted to the topical issue for the industrial sector about the basic laws of polymerization using Ziegler-Natta catalysts. It describes the results of assessing the influence of ethanol content in magnesium chloride on the structure, qualitative, and quantitative properties of Ziegler-Natta (ZN) catalysts synthesized on their basis. The authors use a full range of research methods. It has been shown that an increase in ethanol content leads to a sharp increase in the surface area of the catalyst and a decrease in pore size, which in turn increases the activity of the catalyst but reduces its stereospecificity. The work also presents computer calculations. The main advantages of the article are completeness and integrity, as well as clarity and unambiguity in the description of generally known patterns.
2 Remarks and questions:
2.1 Introduction. Lines 43-44. It is unclear what is meant by "different classifications for polyolefin catalysts," as the given references discuss either phosphine-sulfonate palladium catalyst or titanium-magnesium catalysts modified with 3,3-bis(methoxymethyl)pentane. Perhaps in this case, it would be better to use the term "modification" rather than "classification"
2.2 Introduction. Lines 45-47. The references you cited do not support this proposition.
2.3 Introduction. Lines 55-57. Please check the reference and clarify exactly what was meant, as the cited article was referring to the effect of DIBP addition on the structure of magnesium chloride.
2.4 Introduction. Lines 57-61. It is recommended that a link be provided to verify this information.
2.5 Introduction. Lines 62-66. It is not clear what the contradiction in the results you write about is.
2.6 Introduction. Lines 65-66. If there are posts, links to them should be cited. If they do not exist, they should be cited.
2.7 Introduction. Lines 68-72. The objective is stated specifically, but it is not clear how this wording follows from the above introduction. Why is 1-hexene chosen as a co-monomer? Why are only three different variations of the adduct planned to be investigated? Why exactly these molar ratios?
2.8 Results and Discussion. Lines 117-118. What factors affect catalyst particle size and how can the reproducibility of catalyst fractional composition be controlled?
2.9 Polymerization Results. Lines 279-280. When discussing quantitative differences of a parameter, you can indicate how many times they differ, rather than rewriting the table data. The same applies to the discussion in other sections. In many cases, the results indicate a directly proportional dependence of one or another indicator on the ethanol content in the initial adducts.
2.10 Polymerization Results. Lines 291-293. It is a little strange that a link to these studies is given here for the first time, because the molar ratios of magnesium chloride and ethanol in the work [44] are completely similar to those studied in this article. And reference [45] probably needs a patent number.
2.11 Polymerization Results. Lines 287-288. Also section Propylene Polymerization, lines 101-104. The equal sign is not good when describing the conditions for polymerization.
3 Recommendations:
3.1 Figures 9a and 9b could probably be placed next to each other, making them the same size.
4 Typos:
|
Line number |
Typo |
Amendment |
|
37 |
million |
% (please check with the original) |
|
54, 68 et seq. |
MgCl2.nEtOH |
|
|
90 |
There is a lack of capitalization in the sentence “50 mL of TiCl4 was added gradually for 0.5 h in the suspension” |
|
5 Typos in signatures of axes on graphs:
|
Figure |
Typo |
Amendment |
|
4 a, b, c |
Figure 4 a is smaller in size than figures b and c. |
Figures 2a, b and c should be made the same size. |
Suggestions for language improvement:
|
Line number |
Typo |
Amendment |
|
15 |
To do this - this is not an academic language |
For this purpose |
|
19 |
were |
was |
|
34 |
Very large |
|
|
35 |
annual |
yearly |
|
37 |
An annual |
Worldwide annual |
|
39 |
Due to the … |
The industrial application of thermoplastics has long been of… |
|
40 |
academics |
academic researchers |
|
40 |
, and to this end, many |
. Consequently numerous studies |
|
43-44 |
This led to different classifications for polyolefin catalysts in order to produce new advanced polyolefins [4,5]. |
This resulted in the development of various types of catalysts, with the aim of producing advanced polyolefins [4,5]. |
|
116-118 |
three catalysts started from various adduct compositions, but similar particle sizes were synthesized. |
three catalysts were synthesized starting from different adduct compositions but similar particle sizes. |
|
120-122 |
Then, their performance in propylene polymerizations in terms of activity, isotacticity, H2 response, and comonomer incorporation was evaluated as will be discussed later. |
Then, their performance in propylene polymerizations was evaluated in terms of activity, isotacticity, H2 response and comonomer incorporation, as will be discussed later. |
Author Response
Comments and Suggestions for Authors
1 A brief summary:
This article is devoted to the topical issue for the industrial sector about the basic laws of polymerization using Ziegler-Natta catalysts. It describes the results of assessing the influence of ethanol content in magnesium chloride on the structure, qualitative, and quantitative properties of Ziegler-Natta (ZN) catalysts synthesized on their basis. The authors use a full range of research methods. It has been shown that an increase in ethanol content leads to a sharp increase in the surface area of the catalyst and a decrease in pore size, which in turn increases the activity of the catalyst but reduces its stereospecificity. The work also presents computer calculations. The main advantages of the article are completeness and integrity, as well as clarity and unambiguity in the description of generally known patterns.
OUR ANSWER: We appreciate the long summary of our article, highlighting the positive aspects, above all.
2 Remarks and questions:
2.1 Introduction. Lines 43-44. It is unclear what is meant by "different classifications for polyolefin catalysts," as the given references discuss either phosphine-sulfonate palladium catalyst or titanium-magnesium catalysts modified with 3,3-bis(methoxymethyl)pentane. Perhaps in this case, it would be better to use the term "modification" rather than "classification"
OUR ANSWER: We accept the suggestion, since the word “classification” led to a confusing meaning. In addition, the whole paragraph was rephrased because of another concern of another reviewer.
2.2 Introduction. Lines 45-47. The references you cited do not support this proposition.
OUR ANSWER: We have solved this, relating to the former reference 1.
2.3 Introduction. Lines 55-57. Please check the reference and clarify exactly what was meant, as the cited article was referring to the effect of DIBP addition on the structure of magnesium chloride.
OUR ANSWER: In the related reference, the authors mainly assessed the effect of DIBP addition on the structure of magnesium chloride in the MgCl2.EtOH form, so the removal of EtOH content was considered as well. We added a comment accordingly.
2.4 Introduction. Lines 57-61. It is recommended that a link be provided to verify this information.
OUR ANSWER: We referred the past reference 35 here: “Prog. Polym. Sci. 2018, 84, 89-114”.
2.5 Introduction. Lines 62-66. It is not clear what the contradiction in the results you write about is.
OUR ANSWER: The related text was rewritten.
2.6 Introduction. Lines 65-66. If there are posts, links to them should be cited. If they do not exist, they should be cited.
OUR ANSWER: We have added the reference.
2.7 Introduction. Lines 68-72. The objective is stated specifically, but it is not clear how this wording follows from the above introduction. Why is 1-hexene chosen as a co-monomer? Why are only three different variations of the adduct planned to be investigated? Why exactly these molar ratios?
OUR ANSWER: We agree that propylene/ethylene is among the most used combination, but for industry the combination with 1-hexene is especially remarkable. Actually, alkyl chain of long α-olefin comonomer improves the final polymer’s mechanical properties. On the other hand, in ZN assisted olefin polymerizations, α-olefin incorporation is a major challenge, see our recent article at “J. Ind. Eng. Chem. 2022, 116, 359-370.”, which suppresses these catalysts performances. Improving the related catalyst’s features to successfully incorporate long α-olefins was always followed by the researchers. In this regard, we have focused on ethanol content of adduct to see its effect on the incorporation of 1-hexene, as long alkyl side chain containing comonomer.
It was emphasized at Introduction section.
2.8 Results and Discussion. Lines 117-118. What factors affect catalyst particle size and how can the reproducibility of catalyst fractional composition be controlled?
OUR ANSWER: To unravel these issues, a new text was added in Page 3 as the following:
It is worth to mention that the catalyst particle size originates from adducts particle size, which itself is correlated to the solid content during melt quenching, the stirred speed and emulsifier content.
2.9 Polymerization Results. Lines 279-280. When discussing quantitative differences of a parameter, you can indicate how many times they differ, rather than rewriting the table data. The same applies to the discussion in other sections. In many cases, the results indicate a directly proportional dependence of one or another indicator on the ethanol content in the initial adducts.
OUR ANSWER: Considering the reviewer comment, we have revised the related text.
2.10 Polymerization Results. Lines 291-293. It is a little strange that a link to these studies is given here for the first time, because the molar ratios of magnesium chloride and ethanol in the work [44] are completely similar to those studied in this article. And reference [45] probably needs a patent number.
OUR ANSWER: We have corrected the former reference 45: D. Evangelisti, G. Collina, O. Fusco and M. Sacchetti, Magnesium Dichloride-Ethanol Adduct and Catalyst Components Obtained therefrom, US Pat 7087688 B2, 2006.
2.11 Polymerization Results. Lines 287-288. Also section Propylene Polymerization, lines 101-104. The equal sign is not good when describing the conditions for polymerization.
OUR ANSWER: We must apologize because we have not been able to detect the error, unless it was the different format of equal signs and the neighbouring free spaces.
3 Recommendations:
3.1 Figures 9a and 9b could probably be placed next to each other, making them the same size.
OUR ANSWER: We have modified Figure 9 accordingly.
4 Typos:
|
Line number |
Typo |
Amendment |
|
37 |
million |
% (please check with the original) |
|
54, 68 et seq. |
MgCl2.nEtOH |
|
|
90 |
There is a lack of capitalization in the sentence “50 mL of TiCl4 was added gradually for 0.5 h in the suspension” |
|
OUR ANSWER: First we apologize with the first error, that had to be in percentage and not in million tones. Then, we homogenized in all the text: “MgCl2.nEtOH”. However, if the reviewers or the editorial team consider alternative terms like “MgCl2·nEtOH”, we will change it accordingly. We do not know how to add capitals in the sentence of line 90, but we corrected one piece of it.
5 Typos in signatures of axes on graphs:
|
Figure |
Typo |
Amendment |
|
4 a, b, c |
Figure 4 a is smaller in size than figures b and c. |
Figures 2a, b and c should be made the same size. |
Comments on the Quality of English Language
Suggestions for language improvement:
|
Line number |
Typo |
Amendment |
|
15 |
To do this - this is not an academic language |
For this purpose |
|
19 |
were |
was |
|
34 |
Very large |
|
|
35 |
annual |
yearly |
|
37 |
An annual |
Worldwide annual |
|
39 |
Due to the … |
The industrial application of thermoplastics has long been of… |
|
40 |
academics |
academic researchers |
|
40 |
, and to this end, many |
. Consequently numerous studies |
|
43-44 |
This led to different classifications for polyolefin catalysts in order to produce new advanced polyolefins [4,5]. |
This resulted in the development of various types of catalysts, with the aim of producing advanced polyolefins [4,5]. |
|
116-118 |
three catalysts started from various adduct compositions, but similar particle sizes were synthesized. |
three catalysts were synthesized starting from different adduct compositions but similar particle sizes. |
|
120-122 |
Then, their performance in propylene polymerizations in terms of activity, isotacticity, H2 response, and comonomer incorporation was evaluated as will be discussed later. |
Then, their performance in propylene polymerizations was evaluated in terms of activity, isotacticity, H2 response and comonomer incorporation, as will be discussed later. |
Submission Date
OUR ANSWER: We followed all the advice and have to thank him/her for improving in such a detail the manuscript.
Round 2
Reviewer 1 Report
Comments and Suggestions for Authors
The authors have revised their manuscripts according to the comments. I agree to recommend the publication of this manuscript.